



# Scaling methods of leakage correction in GRACE mass change estimates revisited for the complex hydro-climatic setting of the Indus basin

Vasaw Tripathi[1], Andreas Groh[2], Martin Horwath*[2], RAAJ Ramsankaran[1]

[1]Hydro-Remote Sensing Applications (H-RSA) Group, Department of Civil Engineering, Indian Institute of Technology Bombay, Mumbai, Maharashtra, India

[2]Institut für Planetare Geodäsie, Technische Universität Dresden, Germany

*Correspondence to:* Martin Horwath (Martin.Horwath@tu-dresden.de)

**Abstract.** Total Water Storage Change (TWSC) reflects the balance of all water fluxes in a hydrological system. The Gravity
Recovery and Climate Experiment/Follow-On (GRACE/GRACE-FO) satellite missions are the only means of observing this state variable, distributed as coefficients of a spherical harmonic (SH) model. The well-known correlated noise in these observations requires filtering, scattering the actual mass changes from their true locations. This effect is known as leakage. This study explores the traditional basin and grid scaling approaches and develops a novel frequency-dependent scaling for leakage correction of GRACE TWSC in a unique, basin-specific assessment for the Indus basin. We harness the characteristics
of significant heterogeneity in the Indus basin due to climate and human-induced changes to evaluate the physical nature of these scaling schemes. The most recent, WaterGAP Hydrology Model (WGHM v2.2d), with its two variants, standard (without glacier mass changes) and Integrated (with glacier mass changes), is used to derive scaling factors. For the first time, we explicitly show the effect of inclusion or exclusion of glacier mass changes in the model on the gridded scaling factors. Frequency dependant scaling factors, as a novelty, allow us to compare the differences of scaling seasonal and trend
components separately. We employ time series analysis using the Lomb-Scargle periodogram to decompose the time series and visualize the effect of filtering on different time scales. We find that for the Indus basin, where mass changes of different frequencies are localized, frequency-dependent scaling factors for the entire basin perform similar to the gridded scaling factors, while both outperform basin scaling. The property of frequency-dependent scaling to keep the noise level unscaled can be extremely useful for applications requiring a high signal-to-noise ratio of TWSC observations. Apart from these novel
developments and insights into the traditional scaling approach, our study encourages the regional scale users to conduct specific assessments for their basin of interest.

## 1 Introduction

The terrestrial water storage (TWS) includes all water components on and underneath the Earth's surface, i.e., soil moisture, surface water, groundwater, snowpack, and the water contained in biomass (Zhang, 2017). Regional-scale hydrological studies
mainly deal with terrestrial water storage changes (TWSC) over time. In-situ measurement of TWSC from its components is practically impossible at basin scales. This is due to the inability of current observational methods to include all possible water storage compartments and to the point-scale nature of existing measurements, which does not capture the spatial variability of TWSC in a basin. With the launch of the Gravity Recovery and Climate Experiment (GRACE) satellite mission in 2002, global-scale observation of TWSC was made possible. These observations come at a monthly time scale, making GRACE an
invaluable tool to study seasonal mass changes significant to hydrology (Jiang et al., 2014) and more extended time scales required for climate change studies (Tapley et al., 2019). However, the spatial resolution is of the order of several hundred





kilometers, so GRACE is most accurate and valuable in global or continental scale studies only. The limited spatial resolution is related to errors arising from the measurement process of the GRACE and the modeling deficiencies in the estimation of gravity field parameters. Measurement of inter-satellite ranges along the orbit causes the range rate observations to be more

sensitive to mass changes in this direction resulting in error correlations manifesting as North-South stripes in maps. Satellite altitude (~450 km) and inter-satellite distance (~220 km) cause these errors to be even more prominent at smaller spatial scales, limiting effective spatial resolution. The second source of these errors is the deficiencies and limitations in the background models used in the gravity field parameter estimation, dominated by errors in the atmospheric and oceanic dealiasing models (Kvas et al., 2019).


These errors are addressed by filtering the data, including destriping filters to remove striping and low pass filters to reduce random errors in small spatial scales. Destriping filters followed by low pass filtering is relatively simple, making them attractive for many users and performing well overall (Klees et al., 2008). However, the inevitability of filtering leads to additional uncertainties in TWS estimates arising from signal attenuation and leakage. Leakage errors occur from truncation

of maximum degree in the spherical harmonic model along with additional filtering, which leads to mass changes in the region of interest (ROI) to be affected by mass changes outside the ROI and vice versa.

Signal restoration and leakage correction have been an active area of research in hydro-geodetic communities and have been mainly carried out using three approaches; the additive correction approach, the multiplicative correction approach, and the

scaling factor approach (Long et al., 2015). The scaling factor approach has been the most widely used approach for leakage correction. Its simplicity in application to the gridded TWS products ($1^0$ x $1^0$) popularized and revolutionized the use of GRACE data in the hydrological community. The GRACE Tellus website (Monthly Mass Grids - Land | Get Data – GRACE Tellus, accessed 2019) provides gridded scaling factors calculated from the Community Land Model (CLM4.0) that must be applied to the GRACE grids as a regular procedure. The scaling factor approach (Landerer and Swenson, 2012.) uses simulated

TWSC from Global Hydrology Models (GHM) or Land Surface Models (LSM) and processes them in the same way as GRACE to obtain filtered simulations. A scaling factor is obtained through the least-squares fit between original and filtered model simulations, which is multiplied to GRACE estimates to account for leakage.

Scaling factors have been explored in multiple schemes. Spatially, they can either be lumped, i.e., a single scaling factor for

the basin, or distributed, i.e., gridded at a specific resolution. A time scale-dependent scheme uses different scaling factors for mass changes occurring at different time scales. While basin scaling and gridded scaling are the most popular schemes due to their simplicity and globally acceptable performance, the time scale dependant approach is far less studied. The time scale dependant (or frequency dependant) scaling has been suggested on an application basis in previous studies (Rodell et al., 2009; Landerer and Swenson, 2012; Velicogna and Wahr, 2013) but has not been adequately assessed as a scaling method along

with traditional methods. Hsu and Velicogna (2017) used a different scaling factor for seasonal and trend components at each grid cell for determining the land water storage contribution to sea-level change. However, the properties of such a scaling were not discussed further. In any case, it is established that a single scheme cannot be ascertained to perform well for all regions (Long et al., 2015). In the absence of in-situ observations of TWSC, no single criterion for comparison and performance assessment of these different schemes can be utilized. Hence, a region-specific comparison and performance evaluation is the

ideal way to establish the benefits and drawbacks of each of these schemes.

The sensitivity of the scaling factor approach to the choice of LSM or GHM has been established in numerous studies by deriving and comparing scaling factors from different models (Huang et al., 2019). This sensitivity arises from the difference in underlying model physics, water storage compartments modelled, and the accuracy of forcing datasets used in these models.





Most LSMs and GHMs do not model the effect of human intervention on water storage changes. These human interventions include irrigation water use, groundwater use, and artificial reservoir storage and affect the distribution of TWS changes through complex feedbacks between climate-induced and human-induced changes (Döll et al., 2003). The Water Global Assessment and Prognosis (WaterGAP) hydrology model (WGHM) is one such model that accounts for the human intervention in modelling TWS changes (Döll et al., 2003; Müller Schmied et al., 2016).


Moreover, an integrated version of the WGHM and Global Glacier Model (GGM) (Marzeion et al., 2012) has been recently produced (Cáceres et al., 2020), which includes glacier mass changes in the modelled TWSC. Hence, using models that include these complex interactions in deriving the scaling factors promises a more effective leakage correction in GRACE estimates. Comparing gridded scaling factors from two versions allows us to quantify the change in the spatial distribution of the factors 90    when the model excludes a water storage compartment. Indus basin is chosen as the study region due to its complex hydro-climatic nature, which will provide an opportunity to explore the different scaling schemes and two versions of the model with and without the glacier mass changes.

The objectives of this study are, 1) to present simple processing of GRACE level 2 data to obtain TWS changes for Indus 95    basin, 2) to quantify the amount of leakage error and develop scaling factors under three different schemes, and 3) to evaluate the schemes using residual leakage and scaled GRACE noise levels. Through these objectives, we highlight the need for a basin-specific assessment, develop a novel frequency dependant scheme, show the effect of including or excluding a crucial water storage component in the model used for deriving scaling factors.

## 2 Materials and Methods

### 2.1 Study Area

The Indus River Basin, shown in Fig. 1 (basin boundaries by ICIMOD (2021)), covers an area of 1.14 million km$^2$. The basin spans over four nations, India, Afghanistan, Pakistan, and China, and supports over 215 million people with an approximate water availability of 1,329 m$^3$ per head (Frenken, 2012). The Indus River is a perennial river originating from the Bokar Chu glacier in Mt. Kailash in Tibet. It flows through the Ladakh region in Kashmir, through the glaciers of the Himalayas, 105    Karakoram, and the Hindu Kush ranges, entering the plains of Punjab in Kalabagh, Pakistan. In its journey, it is joined by the Kabul River from the west and Panjnad (Jhelum, Ravi, Chenab, Sutlej, and Beas) from the east to drain into the Arabian sea finally. Almost 50% of its discharge is due to snowmelt (Shrestha et al., 2019). Due to the complex terrain conditions, this basin is a data-scarce basin which makes monitoring of hydrological variables sparse and inconsistent. Being a transboundary basin, cooperation between nations also aggravates this situation. Hence, the TWSC estimates from GRACE being the only 110    source for the entire Indus basin, their uncertainty due to leakage and possible correction approaches must be studied thoroughly.

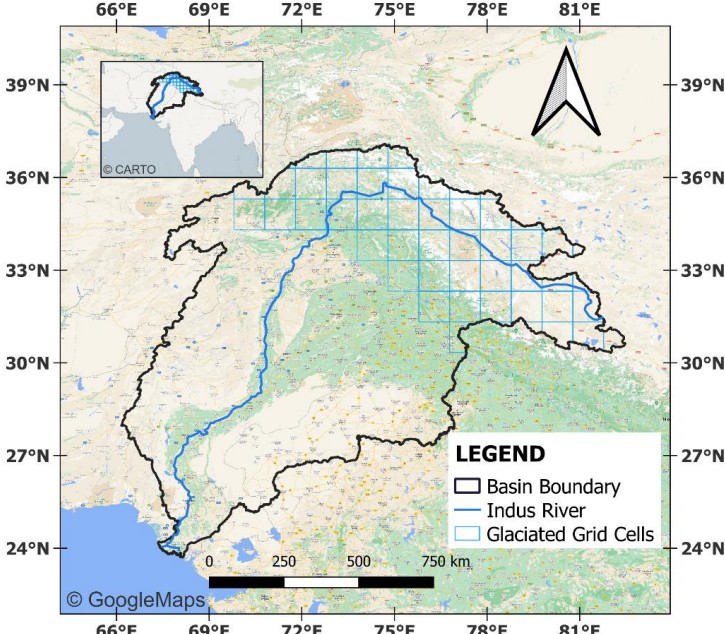

**Figure 1 The Indus Basin. The blue squares represent the glaciated areas as cells of $1^0$ size. Basin boundaries by ICIMOD. Main background map ©Google Maps and inset map ©Carto.**

The winter climate over the Indus basin is dominated by western disturbances embedded with the Indian winter monsoon. During the summer season, the Indian summer monsoon (ISM) brings in precipitation over southern parts of the basin (Dimri et al., 2019). The mean annual rainfall varies between 90 and 500 mm in the downstream and midstream segments, while more than 1000 mm in the upstream catchments. The climate in the Indus Basin varies from subtropical arid and semi-arid to temperate sub-humid in Sindh and Punjab to alpine in the mountainous highlands in the north, with average temperatures

ranging between 2°C and 49°C (Dimri et al., 2019). The hydro-climatic parameters in the Upper Indus basin are primarily influenced by glacier melt and in the lower part by human intervention in the form of different irrigation schemes and extensive groundwater depletion (Rodell et al., 2009).

Hence, a complex hydrological regime exists in the entirety of the Indus basin due to rapidly varying climatological and

anthropogenic conditions across the basin. Figure 2, obtained from the Copernicus Global Land Cover Service, shows the land cover distribution in the Indus basin for 2016. Heavily irrigated areas lie along the river course and central-east region of the basin (Cheema and Bastiaanssen, 2010). Due to the reduction in precipitation and increase in demand, groundwater has been the primary source of irrigation. Due to rising mean annual temperatures, the ET is also high in these regions. Hence, human intervention is the dominant short-scale driver of TWS changes in these regions, as opposed to the upper Indus basin, where

glaciers and snowmelt dominate the TWS changes.



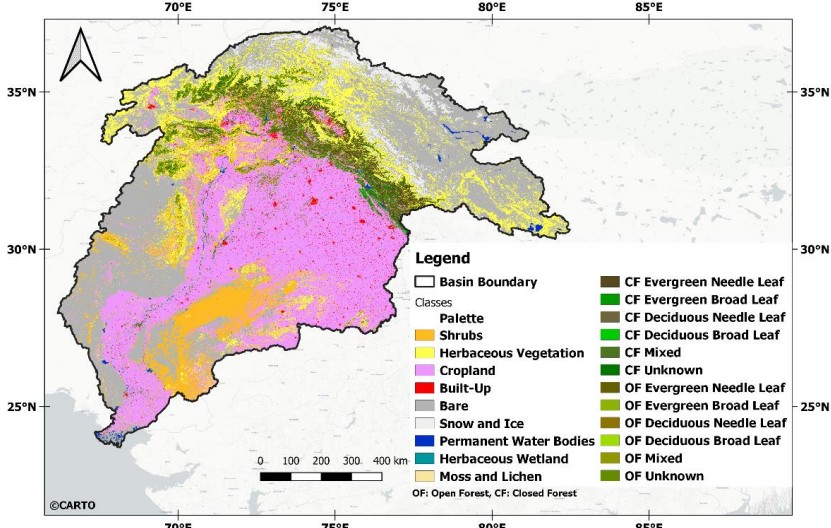

**Figure 2 The land use land cover map (100 m resolution) of the Indus basin for 2016, obtained from the Copernicus Global Land Cover Service. Background map ©CARTO.**

## 2.2 Datasets

### 2.2.1 GRACE Data

The GRACE Level 2 data consisting of monthly Stokes coefficients of Earth's geopotential was used to derive estimates of TWSC over the Indus Basin. The data from three Science Data Centers, Jet Propulsion Laboratory (JPL), University of Texas, Centre for Science and Research (CSR), and GeoForschungsZentrum (GFZ) in release 6 (RL06) were utilized up to degree 90. Only the months common to all three solutions and WGHM output were used. This resulted in 147 monthly solutions from April 2002 to December 2016. The GSM (denoted as is) products were used, containing fully normalized geopotential coefficients representing the full magnitude of land hydrology, ice, and solid Earth processes. In addition, the RL06 MASCON product from CSR was also utilized to compare the overall results of level 2 processing. The ICE-6G_D model was used to account for Glacial Isostatic Adjustment (Richard Peltier et al., 2018). This model is available up to degree and order 256 and provides secular crustal uplift rates in mm/year. The degree 1 coefficients from (Sun et al., 2016), distributed as Technical Note (TN-13) on the Tellus website, were used. The replacement C20 coefficients from (Cheng and Ries, 2017), distributed as TN-11 on the Tellus website, were used.

### 2.2.2 Model Data

TWS anomalies from WGHM 2.2d, both standard and integrated versions, were used (Döll et al., 2003; Müller Schmied et al., 2016; Cáceres et al., 2020). The integrated version simulates glacier mass changes, unlike the standard version. Grids of $0.5^0$ resolution from April 2002 to December 2016 were used and resampled to $1^0$ resolution. For each version, there were four variants available with two climate forcings; ERA-Interim reanalysis (WFDEI) applied with WATCH Forcing Data (WFD) methodology, and bias-corrected using precipitation data sets derived from rain gauge observations of either GPCC v5/v6 (Global Precipitation Climatology Centre, (Schneider et al., 2011)) or CRU TS3.10/TS3.21 (Climate Research Unit, (Harris et al., 2014)]) and two irrigation variants; irrigation water use at 70% of Consumptive Use (CU) and optimal CU. For the Indus basin, no single irrigation scenario can be assigned due to the heterogeneity in irrigation patterns. Hence, an average of these





four variants was taken under each version (Standard and Integrated) to obtain the respective grids. Table 1 summarizes the variants of the model used in this study.

WGHM simulates water flows at a daily time scale on a $0.5^0$ by $0.5^0$ global grid excluding Antarctica. Human intervention and its effect on water flows are also considered by simulating human water use in five sectors: irrigation, manufacturing, livestock farming, domestic use, and thermal power plants. The model requires daily meteorologic inputs of near-surface air temperature, precipitation (rainfall and snow), downward shortwave radiation, and downward long-wave radiation. Reservoir data comes from the Global Reservoir and Dam (GRanD) database, which includes 6862 reservoirs with a total storage capacity of 6197 km³ (Lehner et al., 2011).

Water balance is carried among three water storage compartments in the vertical direction: canopy, snow, and soil water storage. WaterGAP represents soil as a one-layer soil water storage compartment characterized by a land-cover and soil-specific maximum storage capacity and soil texture. The groundwater is recharged from the soil, and storage is simulated after accounting for net abstractions from groundwater due to human use. A fraction of groundwater recharge is assumed to flow back to surface water bodies, and groundwater recharge under surface water bodies is neglected. The surface water storage comprises five sub-compartments: local lakes, local wetlands, global lakes, reservoirs, and global wetlands. At each stage, the storage is simulated by accounting for ET losses and net abstractions from lakes and reservoirs. Finally, the output from surface water bodies and groundwater flows into rivers, where the river water storage is simulated after accounting for the streamflow. Groundwater-Surface Water Use (GWSUSE) model accounts for net abstractions from surface water and groundwater. Irrigation is assumed to vary with countries (efficiency of irrigation water use) and source of irrigation water. In groundwater-depleted areas, the CU from irrigation is considered 70% of the optimal CU since farmers have less availability to satisfy the optimal requirements. In all other areas, it is assumed to be optimal. Country-specific efficiency values are used for surface water irrigation, while in case of groundwater irrigation, water use efficiency is set to a relatively high value of 0.7 worldwide. These abstractions are accounted for in the water balance for each compartment as described above.

The glacier mass changes not included in the WGHM are obtained from Global Glacier Model (GGM) (Marzeion et al., 2012). GGM consists of a surface mass balance model based on the temperature index approach and model that accounts for changes in glacier geometry in feedback to compute the changes in glacier mass. The model is forced by a mean ensemble of seven atmospheric datasets, reducing the uncertainty inherent in the individual forcing datasets. As an initial condition for glacier geometry, data like glacier areas and boundaries are taken from Randolph Glacier Inventory (RGI v6) (Pfeffer et al., 2014). The model is calibrated using observed surface mass balance from World Glacier Monitoring Service (WGMS 2016).





Table 1 Summary of WGHM 2.2d versions used in this study

| Model Version | Precipitation Bias | Consumptive Irrigation Water Use | Model Name |
|---|---|---|---|
| Standard WGHM (std) | GPCC | 100% | WaterGAP22d_std_WFDEI_GPCC_mm_irr100 |
| | GPCC | 70% | WaterGAP22d_std_WFDEI_GPCC_mm_irr70 |
| | CRU | 100% | WaterGAP22d_std_WFDEI_CRU_mm_irr100 |
| | CRU | 70% | WaterGAP22d_std_WFDEI_CRU_mm_irr70 |
| Integrated WGHM (gl) | GPCC | 100% | WaterGAP22d_gl_WFDEI_GPCC_mm_irr100 |
| | GPCC | 70% | WaterGAP22d_gl_WFDEI_GPCC_mm_irr70 |
| | CRU | 100% | WaterGAP22d_gl_WFDEI_CRU_mm_irr100 |
| | CRU | 70% | WaterGAP22d_gl_WFDEI_CRU_mm_irr70 |

## 2.3 Methods

### 2.3.1 GRACE Data Processing

The GRACE level 2 SH coefficients from the GSM product represent the geopotential of Earth containing the full magnitude of land hydrology, ice, and solid Earth processes during each month *t* as in eq. 1,

$$V(r,\theta,\lambda,t) = \frac{GM}{R_e}\sum_{n=0}^{N}\left(\frac{R_e}{r}\right)^{n+1}\sum_{m=0}^{n}P_n^m(cos\,\theta)\{C_{nm}^t cos\,m\lambda + S_{nm}^t sin\,m\lambda\} \tag{1}$$

where $\theta,\lambda$ are co-latitude and longitude respectively, *n,m* are degree and order of the expansion, *N* is the maximum degree used which was 90 in this case, $P_n^m$ are the associated Legendre's functions and $C_{nm}^t$ and $S_{nm}^t$ are the spherical harmonic coefficients at month *t*, $R_e$ (6378 km) is Earth's equatorial radius.


The mean static gravity field is removed to obtain gravity field anomalies. In this study, the mean period was taken as the mean of the entire study period. Monthly solutions common to all three (JPL, CSR, and GFZ) centres till December 2016 were extracted with a tolerance of 15 days in the middle of each monthly solution epoch. Running means over three consecutive solutions are taken, which reduces the noise in the solutions. Glacial Isostatic Adjustment (GIA) is the ongoing secular response of Earth's surface to the mass changes that occurred due to the deglaciation process on Earth and is removed using the ICE-6G model (Richard Peltier et al., 2018). The same GIA model used in CSR Mascons is used to maintain consistency. Assuming mass changes occurring in a thin spherical shell (~15 km) around Earth (Wahr et al., 1998), these coefficients are multiplied by degree-dependent factors to obtain surface mass changes coefficients in terms of equivalent water height (EWH). The resulting change in surface mass density is obtained as eq. 2,

$$\Delta\sigma(\theta,\lambda,t) = \frac{R_e\rho_{avg}}{3}\sum_{n=0}^{N}\left(\frac{2n+1}{1+k_n'}\right)\sum_{m=0}^{n}P_n^m(cos\,\theta)\{\Delta C_{nm}^t cos\,m\lambda + \Delta S_{nm}^t sin\,m\lambda\} \tag{2}$$

where $\rho_{avg}$ is the average density of Earth (5515 kg/m³), $k_n'$ are the elastic load Love numbers of Earth.

Swenson Filter (Swenson and Wahr, 2006) is used for destriping, which removes the correlated errors in even and odd degrees. The filter parameters are chosen after (Sasgen et al., 2018), which minimizes the signal and noise contamination between mid-latitude and polar regions during destriping. A Gaussian low pass filter of half width 300 km is employed to reduce the random errors in higher degree terms. Grids of filtered surface mass changes are obtained as eq. 3,

$$\Delta\sigma(\theta,\lambda,t) = \frac{R_e\rho_{avg}}{3}\sum_{n=0}^{N}\left(\frac{2n+1}{1+k_n'}\right)\sum_{m=0}^{n}P_n^m(cos\,\theta)\,W_n\{\Delta C_{nm}^t cos\,m\lambda + \Delta S_{nm}^t sin\,m\lambda\} \tag{3}$$





where $W_n$ are degree-dependent factors of Gaussian filter in spectral-domain (Sasgen et al., 2018).

From the filtered GRACE surface density coefficients, mass changes for the Indus basin are obtained by regional integration using the region function defined in eq. 4:

$$RF(\theta, \lambda) = \begin{Bmatrix} 1 \\ 0 \end{Bmatrix} \begin{matrix} \forall \ (\theta, \lambda) \in R \\ \forall \ (\theta, \lambda) \in \Omega - R \end{matrix} \tag{4}$$

where $RF$ is the region function inside the region $R$ of Indus basin and region $\Omega$ is the total surface area of Earth. This region function is transformed to spectral-domain (up to degree 90), and the corresponding coefficients are multiplied with filtered

surface density coefficients to obtain mass changes for Indus basin, $\Delta M$ as in eq. 5,

$$\Delta M(\theta, \lambda, t) = \frac{R_e \rho_{avg}}{3} \sum_{n=0}^{N} \left( \frac{2n+1}{1+k'_n} \right) \sum_{m=0}^{n} P_n^m (\cos \theta) W_n \{ R_c \Delta C_{nm}^t \cos m\lambda + R_s \Delta S_{nm}^t \sin m\lambda \} \tag{5}$$

where $R_c$ and $R_s$ are respectively the cosine and sine coefficients of the region function in the spectral domain.

### 2.3.2 Scaling Factors Determination

WGHM anomalies were centred around the mean of the entire observation period (April 2002-December 2016) by including

only GRACE months. Unfiltered mass change time series for the Indus basin from both versions were obtained using the region function in eq. 4, by integrating the global grids over the surface of Earth as in eq. 6,

$$\Delta M_{region}(t) = \frac{1}{\Omega_{region}} \int \Delta\sigma(\theta, \lambda, t) \, RF(\theta, \lambda) d\Omega \tag{6}$$

where $d\Omega = R_e^2 \sin\theta d\theta d\lambda$ is an element of the area on the region surface and $\Omega_{region}$ is the total area of the region. The global grids are then transformed to the spherical harmonic domain up to degree 90 and applied with Swenson destriping and Gaussian

Low pass filter in the same manner as described in Sect. 2.3.1. The filtered coefficients are transformed back to the spatial domain to obtain filtered model grids. Regional integration in the spectral domain is performed with filtered model coefficients to get filtered model mass change time series.

A single scaling factor ($k$) for the entire basin is derived by a least-squares regression between unfiltered model time series

and filtered model time series, that is, by minimizing $L$ in eq. 7,

$$L = \sum_{t=1}^{T} (\Delta S_t^u - k\Delta S_t^f)^2 \tag{7}$$

over the entire period of study. In eq. 7, $\Delta S_t^u$ is the unfiltered and $\Delta S_t^f$ is the filtered model mass for the $t^{th}$ month. A scaling factor for each grid cell in a $1^0$ x $1^0$ grid is derived using the unfiltered model grid and filtered model grid and minimizing $L$ in eq. 7 but at every grid cell in the basin. This is done for both versions of WGHM to obtain a $1^0$ resolution map of scaling

factors.

To derive frequency-dependent scaling factors, the total mass changes from unfiltered and filtered model versions are decomposed into long term linear, seasonal and residual components as eq. 8,

$$\Delta S = \Delta S_{long \ term} + \Delta S_{sesonal} + \Delta S_{residual} \tag{8}$$

For this purpose, a time series model is fit to the data using least squares as in eq. 9,

$$y_i = \beta_0 + \beta_1(t) + \sum_{k=1}^{n} \left[ \beta_k^c \cos\left( \frac{2\pi}{T_k} t \right) + \beta_k^s \sin\left( \frac{2\pi}{T_k} t \right) \right] + v(t) \tag{9}$$

where $\beta_1$ is the linear trend, $\beta_k^c$ and $\beta_k^s$ are the amplitudes of the cosine and sine terms of the periodic component with a period $T_k$ and $t$ represents the month w.r.t the mean of the observation period. However, the periods ($T_k$) of the seasonal terms are unknown. For this study, the unknown periods are found from the data itself using a Lomb-Scargle (LS) Periodogram analysis

(Scargle, 1982) which allows detection of weak periodic signals in otherwise random, unevenly sampled data.





From this analysis, the peaks were found at annual and semi-annual periods. The false alarm probabilities associated with each peak were minimal ($\sim 10^{-5}$), which shows that these periods are significant. These periods were used in eq. 9 for time series decomposition. The trend, annual, and semi-annual components are separated in both model versions' unfiltered and filtered time series. Then a least-squares fit is carried out for all three components separately to obtain three scaling factors that minimize the expression in eq. 10,

$$L = \sum_{j=1}^{N}(\Delta S_j^u - k_j\, \Delta S_j^f) \ \text{where } j = \left\{\begin{array}{c} \text{trend} \\ \text{annual} \\ \text{semi} - \text{annual} \end{array}\right\} \tag{10}$$

where $k_j$ is the scaling factor for $j^{th}$ component.

Figure 3 presents the schematic of deriving scaling factors (in green boxes) from filtered and unfiltered model time series (in yellow boxes) and applying them to corresponding GRACE time series (in blue boxes) to obtain scaled GRACE estimates (in purple boxes).

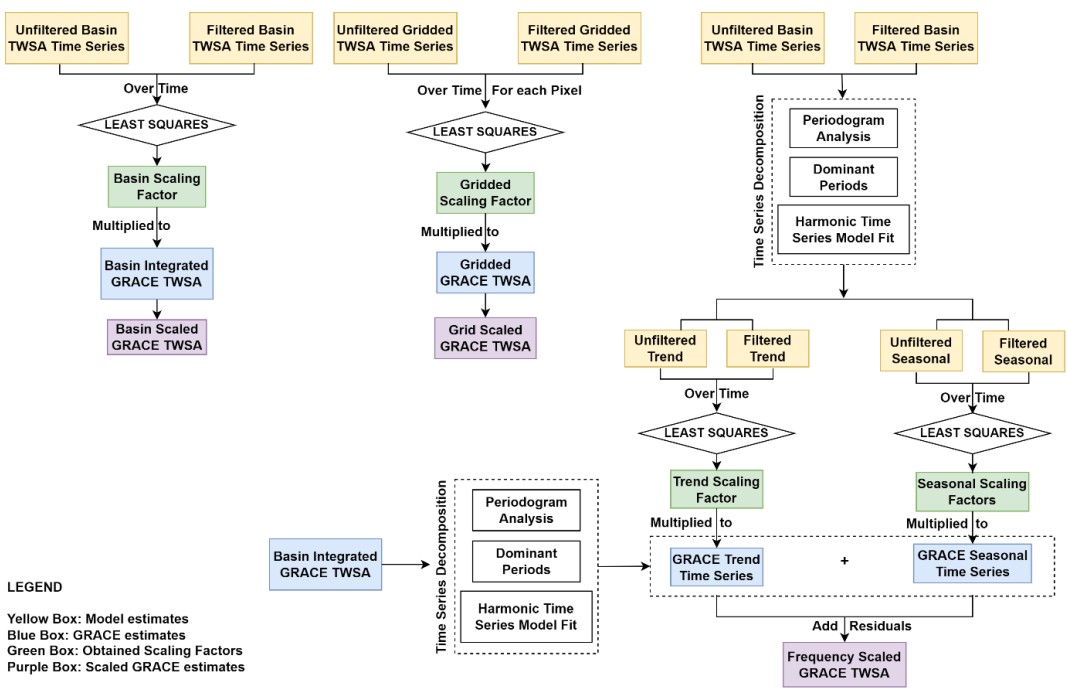

**Figure 3 The schematic diagram for deriving and applying the three schemes of scaling.**

### 2.3.3 Assessment of Uncertainty Components

Two quantities are computed using WGHM as a proxy to the true signal to assess the uncertainty specific to leakage and scaling. Equation 11 provides the initial leakage due to filtering, and equation 12 provides the amount of leakage that is not covered by the scaling factors, termed residual leakage.

$$Initial\ Leakage\ Error = \text{std}(\Delta S_t^u - \Delta S_t^f)\ x\ \frac{\text{RMS}(\Delta G_t)}{\text{RMS}(\Delta S_t^f)} \tag{11}$$

$$Residual\ Leakage\ Error = \text{std}(\Delta S_t^u - k\Delta S_t^f)\ x\ \frac{\text{RMS}(\Delta G_t)}{\text{RMS}(\Delta S_t^f)} \tag{12}$$





where std represents the standard deviation, and RMS represents the root mean square of the time series indexed by $t$, $k$ represents any one of the scaling factors schemes, $\Delta G_t$ represents GRACE mass time series. The residual leakage should be less than the initial leakage if the scaling factors work. This quantification assumes all the variation in the difference of filtered

and unfiltered model time series to represent leakage error (making no distinction between individual effects on underlying signal and noise). It is thus meaningful for intercomparison of different schemes but not as a true estimate of leakage error. This standard deviation is multiplied by the ratio of RMS of GRACE and model time series to account for the amplitude difference in GRACE and model (Landerer and Swenson, 2012).

The uncertainty estimation of GRACE mass estimates is done following the approach in (Groh et al., 2019), which uses only the time series of derived GRACE masses to determine noise. The unscaled GRACE time series contains errors from measurement, leakage, GIA, and C20. Since this approach provides the temporally uncorrelated component of total GRACE errors, the effect of scaling on the random component of time series (thus a random component of leakage) can be compared. The uncertainties from GIA models (<1mm/y EWH) and $C_{20}$ (<0.1 mm/y) coefficients are almost negligible for the Indus basin

and hence, are not considered (Caron et al., 2018; Blazquez et al., 2018). A high pass filter is applied to the residuals of the GRACE time series (after removing trend and seasonal components) to remove the unmodeled inter-annual components in the residuals. The filter width is chosen to be 18 months (6 sigma width of Gaussian hat=18 months), which means all the signal with a larger than 18-month period is removed, leaving temporally uncorrelated errors referred to as noise. This noise contains the GRACE measurement error and the random component of the initial leakage error. A scaling factor generated from high

pass filtering random white noise signals is multiplied to account for the damping of any white noise component during high pass filtering. Similarly, this process is repeated for scaled GRACE time series. The corresponding scaled noise contains the scaled measurement and residual random leakage errors.

## 3 Results and Discussion

### 3.1 Mass Change Estimates from GRACE and WGHM

Figure 4 shows the result of the processing scheme followed to obtain mass change time series from the average of three GRACE level 2 SH solutions and MASCON time series from CSR. The shaded region depicts the 3σ interval of uncertainties from formal errors of the SH coefficients provided by the centers. These uncertainties are propagated to the mass estimates and averaged for three centers by the law of error propagation over equation 5 in Sect. 2.3.1. A GRACE trend value of -7.6 ± 0.6 Gt/year is obtained (Table 2). Similar values of -8.6 Gt/y from (Scanlon et al., 2018) and -9.1 Gt/y from (Kvas et al., 2018)

have been reported. The differences can be attributed to processing strategies and different data releases used in these studies. The performance of the processing scheme is assessed by comparison to the time series obtained from the Level 3 Release 6 CSR MASCON (CSR-M) solution (Save et al., 2016). Although derived from the same level 1 data as SH solutions, MASCONS are constrained to reduce the leakage effect arising from the post-processing step. Figure 4 also shows the GRACE SH and CSR-M time-series correlation for the Indus basin. The high value of Pearson's correlation coefficient ($R^2 = 0.95$)

indicates that the SH solution and processing strategy used are nearly as good as the MASCON solution for the Indus basin. An $R^2$ value of 0.91 is obtained when both time series are de-trended, which confirms that the high values are not a spurious result.

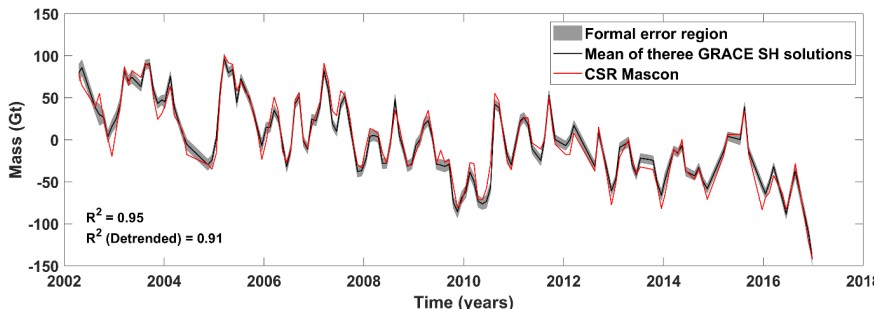

**Figure 4 Time Series of Mass anomalies (mean removed: 2002-2016) in the Indus basin derived from GRACE Spherical Harmonic**
**solution (mean of JPL, CSR, and GFZ), corrected with ICE6G and filtered with Swenson destriping and 300 km Gaussian filter.**

Figure 5 shows the mass anomaly time series from the standard and integrated version of WGHM along with the GRACE SH-based time series. A significantly more negative trend is seen in the integrated version (-20.5 ± 0.4 Gt/y) than in the standard version (-10.3 ± 0.4 Gt/y), indicative of significant glacier melt in the Indus basin, which has a dominating effect on the overall TWS trend of the Indus basin [35]. The negative trend from the standard version can be attributed to the increasing losses from
ET due to climate-induced changes from increasing mean annual temperatures (Shrestha et al., 2019) and human-induced changes due to severe groundwater depletion for meeting the irrigation demands in the Indus plains (Rodell et al., 2009).

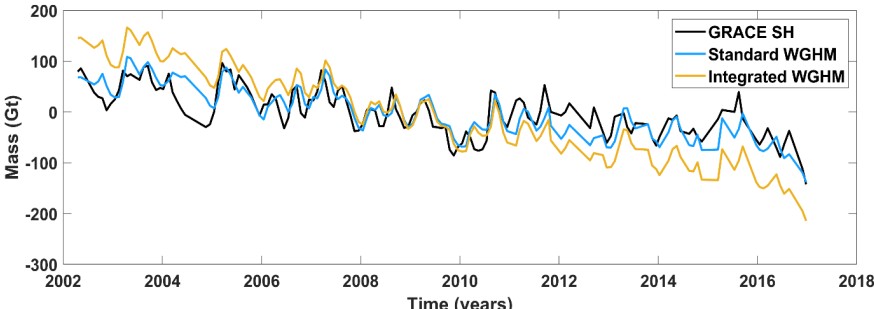

**Figure 5 Comparison of time series from GRACE SH, Standard WGHM, and Integrated WGHM.**

The periodogram analysis of the standard and integrated version (Fig. 6(b) and 6(c)) shows clear peaks at annual and semi-annual frequencies. The peak frequencies obtained from the model and GRACE (Fig. 6a) are almost identical, confirming the ability of GRACE to observe small seasonal signals in the Indus basin, which is mainly dominated by trends. However, from subsequent time-series fit (Table 2), the annual amplitude (10.8±3 Gt) obtained from GRACE is much smaller than the semi-annual amplitude (23.8±2.9 Gt). It is found to be the artifact of filtering, as explained later in Sect. 3.2. Figure 7 shows the
spatial distribution of different components of mass changes from WGHM Standard and Integrated versions. Close to the northern basin boundary, the annual signal content is higher in the integrated version than in the standard version, while the semi-annual component is almost the same. This is due to the annual component of glacier mass change contribution absent in the standard version. Respective amplitudes from the model time series fit (Table 2) also reflect this. The semi-annual seasonality in TWS changes results from bimodal precipitation distribution in the Indus basin. The inter-annual signal content
(area of the frequency spectrum in the inter-annual bandwidth) is almost identical in both model versions and GRACE. This shows that the addition of glacier mass changes to WGHM does not contribute to the inter-annual variations. These inter-annual variations are probably the result of long-term groundwater depletion (Pradhan, 2014) and inter-annual variability





resulting from winter/spring precipitation over the upper Indus basin due to El Niño Southern Oscillation (ENSO) (Krakauer, 2019).


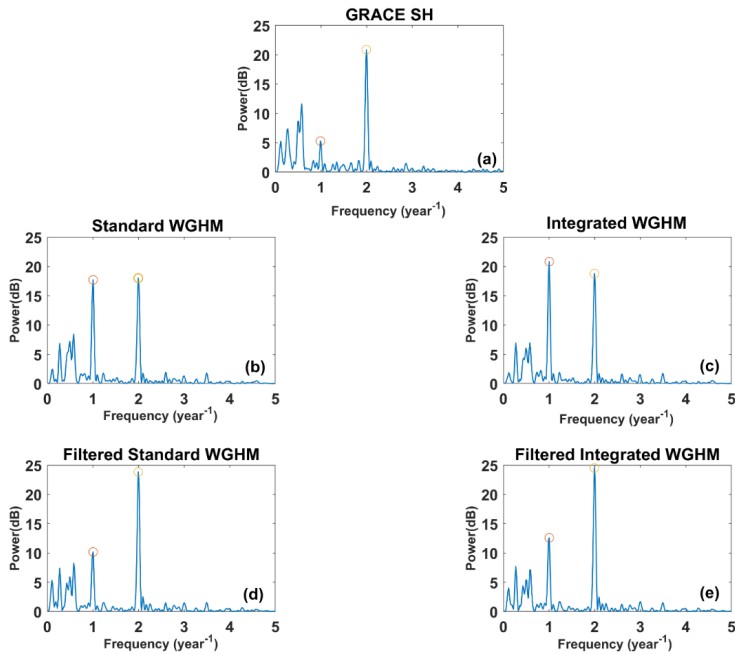

**Figure 6 (a) Periodogram for GRACE (b) for Unfiltered Standard Model (c) for unfiltered Integrated (d) for filtered standard (e) for filtered integrated Model time series. The peaks marked are the ones that are selected for time series fit.**

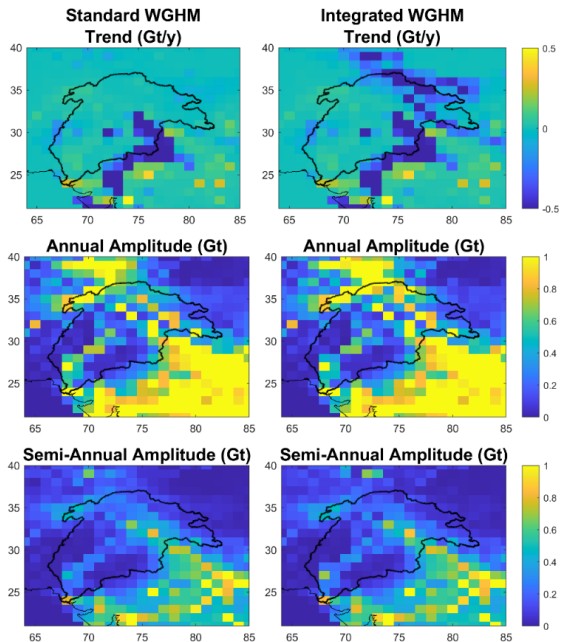


**Figure 7 Trend, Annual, and Semi-Annual Mass changes from WGHM Standard (left panel) and Integrated (right panel) versions.**





**Table 2 Summary of Parameters from Time Series Fitting**

| Time Series | Trend (Gt/year) | Annual Amplitude (Gt) | Semi-Annual Amplitude (Gt) | RMSE (Gt) |
|---|---|---|---|---|
| GRACE | -7.6 ± 0.6 | 10.8 ± 3 | 23.8 ± 2.9 | 27.7 |
| Unfiltered Standard Model | -10.3 ± 0.4 | 17.7 ± 2.3 | 17.7 ± 2.3 | 19.8 |
| Unfiltered Integrated Model | -20.5 ± 0.4 | 19.3 ± 2.2 | 17.9 ± 2.2 | 18.9 |
| Filtered Standard Model | -9.0 ± 0.3 | 10.8 ± 2 | 17.1 ± 2 | 17.1 |
| Filtered Integrated Model | -16.7 ± 0.3 | 12.1 ± 1.9 | 17.2 ± 1.9 | 16.6 |

**3.2 Effect of Filtering**

Spatially, the effect of filtering is visualized in Fig. 8, which shows mean TWS anomalies for each calendar month from the unfiltered and filtered WGHM Integrated version. Local features and small-scale changes in TWS are smoothed out and reduced in amplitude. Large-scale changes in TWS (right panel, Fig. 8) follow the pattern of bimodal precipitation distribution in the Indus basin due to western disturbances and ISM (Hasson et al., 2016; Hussain et al., 2020). The northwest to southeast

increase in storage (reduction of blue colour intensity) during the winter months (December to February) and pre-monsoon months (March to May) can be attributed to Western Disturbances. It is worth mentioning that there is generally a 1-month lag between precipitation and storage in this region. The southeast to the northwest increase of storage (decrease in blue and increase of yellow intensity) during the summer from July to October can be attributed to ISM. October and November show the retreat of monsoon and hence decreasing storage. Small-scale TWS changes (left panel, Fig. 8) are driven by heavy

irrigation along the river, southeast Indus plains, and snow and glacier melt in the upper Indus basin.





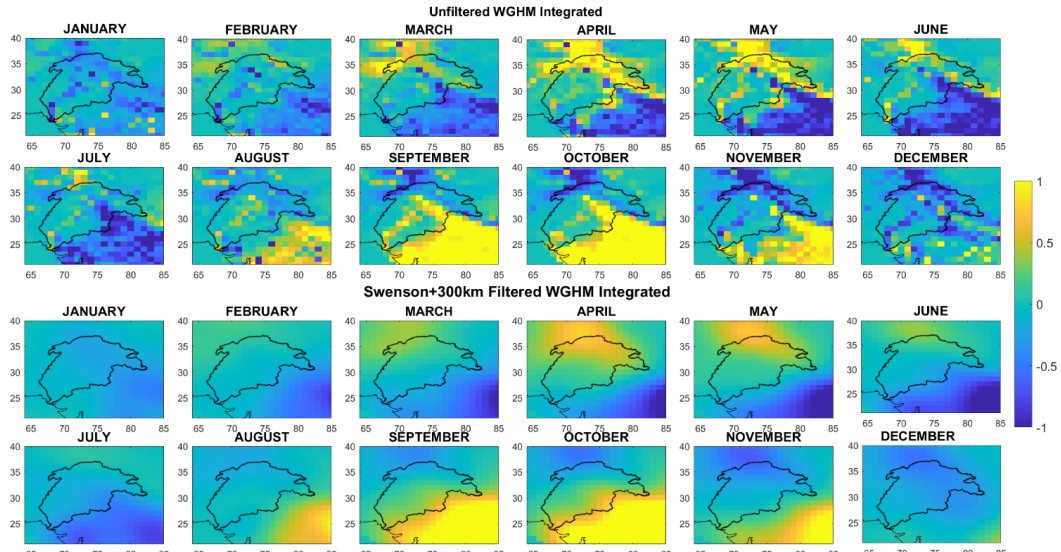

**Figure 8 Mean Anomalies for each month from April 2002-December 2016 obtained from WGHM Integrated: unfiltered (left panel) and filtered (right panel). Refer to Fig. 2 for land cover information.**

Temporally, the filtering dampens the trend, i.e., makes it less negative, in both the model versions (Fig. 9a and 9b). The
dampening is by ~13% for the standard version and stronger, by ~20%, for the integrated version. The filtering brings the model-based trends closer to the GRACE-based trends, partly explaining the trend differences between filtered GRACE-based and unfiltered model-based time series.

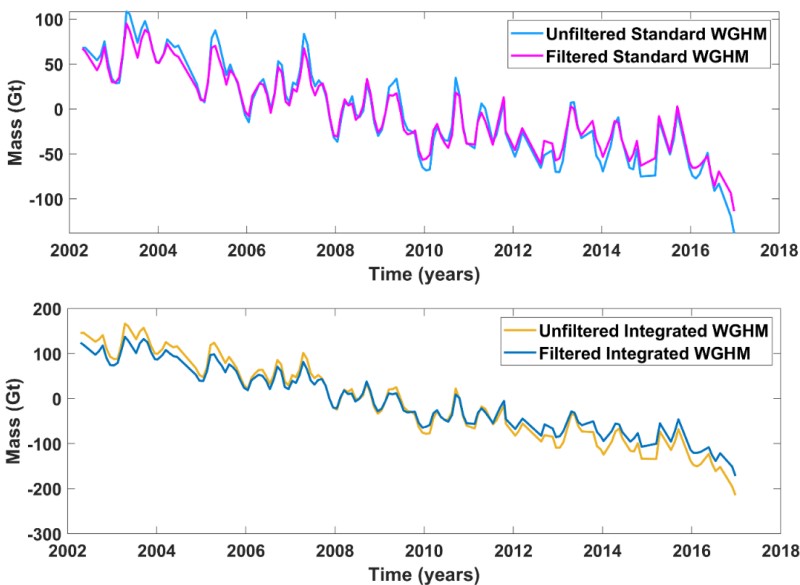

**Figure 9 Unfiltered and Filtered time series from (a) Standard version and (b) Integrated WGHM version. Note the different y-axis scales.**





The effect of filtering on seasonal signals can be analyzed from Figs. 6d and 6e. The peaks are obtained at the same frequencies as the unfiltered time series, which shows that filtering does not induce any additional periodicity of mass changes in the Indus basin. From the relative peak heights of annual and semi-annual frequencies, it is also clear that filtering affects annual and semi-annual mass changes differently. The time series fit (Table 2) shows that filtering significantly reduces the annual amplitudes in both versions, while the semi-annual amplitudes are reduced only slightly. Hence, both versions show annual mass variations to leak out, explaining why the annual peak from GRACE (Fig. 6a) is suppressed. This is a significant effect of filtering and must be restored by scaling.

### 3.3 Scaling Factors

#### 3.3.1 Basin Scale Factors

Basin scaling factors of $k_s=1.14$ from the standard version and $k_g=1.22$ from the integrated version are obtained. These values are greater than one, indicating that filtering causes the overall mass changes to leak out, and hence, a factor greater than one is required to restore the signal. However, the values are small, indicating that the Indus basin has a small leakage amount overall. Basin scale factors from two studies have reported similar values (Table 3). The addition of glacier mass changes in the model leads only to a minor effect on the basin scale factor. Glacier mass changes are located at the edge of the basin and suffer from more leakage out. This explains the slightly larger scaling factor from the integrated version. The similarity of our results to results from other studies that used different models indicates that basin scale factors are not very sensitive to the model used for the Indus basin.

**Table 3 Basin Scale Factors and Frequency-Dependant scaling factors for Indus basin. Basin Scaling factors from two different studies are given for reference.**

| WGHM Models | Basin | | | Frequency-Dependent | | |
|---|---|---|---|---|---|---|
| | This Study | GLDAS NOAH (Landerer and Swenson, 2012) | CLM 2.0 (Long et al., 2015) | $k_{trend}$ | $k_{annual}$ | $k_{semi-annual}$ |
| Standard | 1.14 | 1.34 | 1.12 | 1.14 | 1.64 | 1.02 |
| Integrated | 1.22 | | | 1.23 | 1.59 | 1.03 |

#### 3.3.2 Gridded Scaling Factors

Figure 10 shows the gridded scaling factors in $1^0$ resolution maps from the standard (left) and the integrated version (right). The maps include 108 scaling factors, one for each pixel inside the Indus basin. From the standard version, the scaling factors ranged from -0.4 to 10.1. From the integrated version, the scaling factors ranged from -0.5 to 8.5. Significant differences between the two occur in the upper Indus basin, where the glaciers are located. Table 4 lists the standard interpretation of gridded scaling factor values used in most studies (Long et al., 2015). Figure 11 shows the histogram of scaling factors from both model versions.





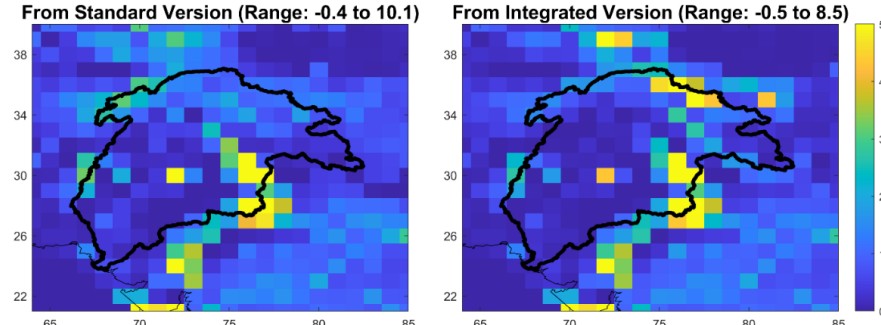

405

**Figure 10 Gridded scaling factor from the standard (left) and Integrated (right) version of WGHM. The grid size is 1⁰ equiangular.**

**Notice the difference lies mainly in glaciated grid cells in the Upper Indus Basin.**

**Table 4 Interpretation of Scaling factors**

| Scaling Factor | Interpretation |
|---|---|
| k<0 | Out of phase |
| 0<k<0.3 | Prominent leakage in |
| 0.3<k<1 | Moderate leakage in; amplitude lower |
| k=1 | No leakage |
| 1<k<3 | Moderate leakage out; higher amplitude |
| k>3 | Significant leakage out |

410

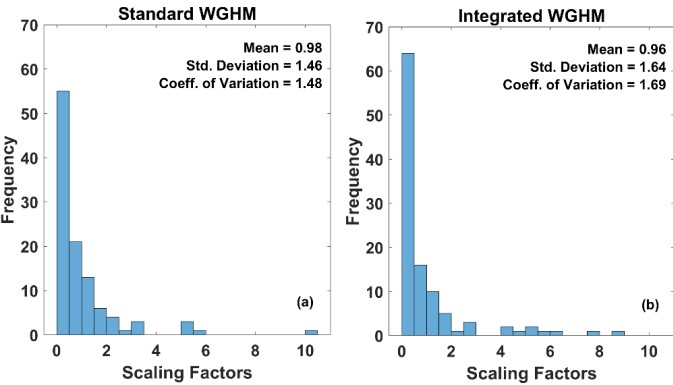

**Figure 11 Frequency distribution of scaling factors from (a) Standard, (c) Integrated version for Indus basin. The mean, standard deviation and Coefficient of Variation (CV) are shown inset. Negative scaling factors are excluded.**





From the standard version, scaling factors for 13 grid cells are negative. These grid cells were excluded for scaling, considering out-of-phase behaviour with respect to their neighbouring grid cells. The majority of scaling factors are less than 1 (67 grid cells), out of which most are less than 0.5 (55 grid cells), indicating that a large part of the basin suffers from moderate to significant leakage-in. These small scaling factors occur mainly in the lower reaches of the basin (Indus plains) and in the upper Indus basin, where non-glacier mass changes are small in magnitude. Few grid cells stand out with larger scaling factors (>3), which depict large local mass changes in the pixel. These occur in the southeast Indus basin region, which has a larger magnitude of TWS changes due to ISM and human intervention in the form of irrigation and groundwater depletion. These more considerable changes leak out to nearby dry regions of the upper Indus basin (as represented in the standard version), requiring greater than one, scaling factor.

From the integrated version, scaling factors at 11 grid cells are found negative and excluded. The number of grid cells with significant leakage in (k<0.5) is larger (62) than for the standard version. The grid cells with large scaling factors (k>3) are more distributed than the standard version. This is because the glacier mass changes cause large local mass variations in the upper Indus basin, requiring larger scaling factors to account for leakage out to nearby dry regions (north-east of the Indus basin with arid Tibetan plateau). The spatial variability of scaling factors obtained is higher (CV = 1.69) from the integrated version than from the standard version (CV = 1.48) due to a more heterogeneous representation of mass changes in the integrated version.

Figure 12 shows the histograms of grid scaling factors only for the non-glaciated region of the Indus basin (i.e., the non-glaciated pixels in Fig. 1). The increase in the frequency of small scaling factors (<0.5) and decrease in large ones (>3) indicates that the distribution of values from the integrated version is shifted towards zero as compared to the standard version. This shows that the addition of a localized water storage compartment in the model (glacier in this case) not only affects the scaling factors in that region but in the entire basin.

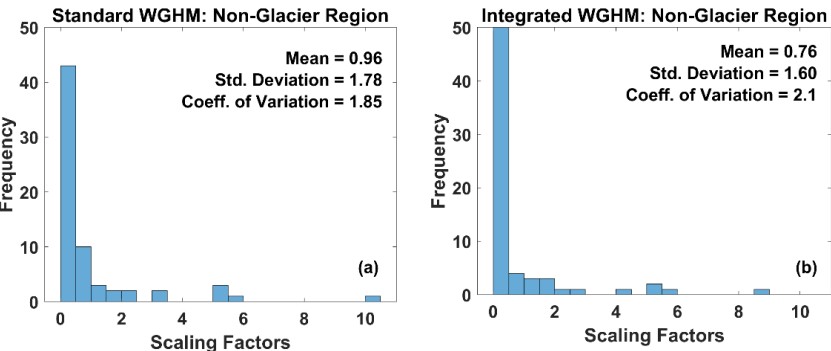

**Figure 12 Histogram of scaling factors from (a) Standard and (b) Integrated version for the non-glaciated region of Indus basin. Notice the increased frequency of scaling factors <0.5 in the integrated version.**

### 3.3.3 Frequency-Dependent Scaling Factors

The frequency-dependent scaling factors for the Indus basin are also shown in Table 3. For the trend component, the scaling factors from both versions (1.14 from Standard and 1.23 from Integrated) are almost identical to the frequency-independent basin scale factors. This reinforces that the Indus basin is dominated by long-term trends rather than seasonal signals, which causes the basin scale factors to be driven by filtering the trend component. The scaling factors for the annual component from both versions (1.64 from standard and 1.59 from integrated) are significantly higher than 1 to account for leakage out of the





annual mass changes described earlier. This also shows that the basin scale factor alone would under-scale the annual component. The semi-annual scaling factors are nearly identical and close to 1 in the standard and integrated version showing that filtering has a negligible effect on this component. These deviations from a single basin scale factor reinforce the need for

frequency-dependent scaling factors in basins where mass changes occur at different frequencies and the filtering has a significantly different effect on these individual frequencies.

### 3.4 Scaled GRACE Mass Estimates

Table 5 shows the scaled GRACE time series parameters from each scaling scheme. Trends become more negative (compared to -7.6 Gt/y from unscaled GRACE) from all three scaling schemes. Similar scaled trends from the basin and frequency-

dependent scaling show the dominance of the trend component in the Indus basin. Grid scaling leads to the most negative trends since grid cells with significant local mass changes contributing to the overall trend are scaled more with larger scaling factors than basin averaged factors. All three scaling schemes restore the annual mass changes lost due to filtering, as evident from increased annual amplitude (compared to 10.8 Gt from unscaled). Grid scaling seems to overestimate the annual amplitude when compared to frequency-dependent scaling. Most grid cells contributing to annual mass changes are also

dominated by significant decreasing trends, leading to larger scaling factors for those grid cells and, hence, larger scaling of annual amplitudes (Fig. 7). Different mass change components from Grid-scaled GRACE from both model versions are shown in Fig. 13. It can be seen that the model (Fig. 7) does not entirely drive the spatial distribution of respective components in scaled GRACE, which also contains the large-scale patterns seen in unscaled GRACE.

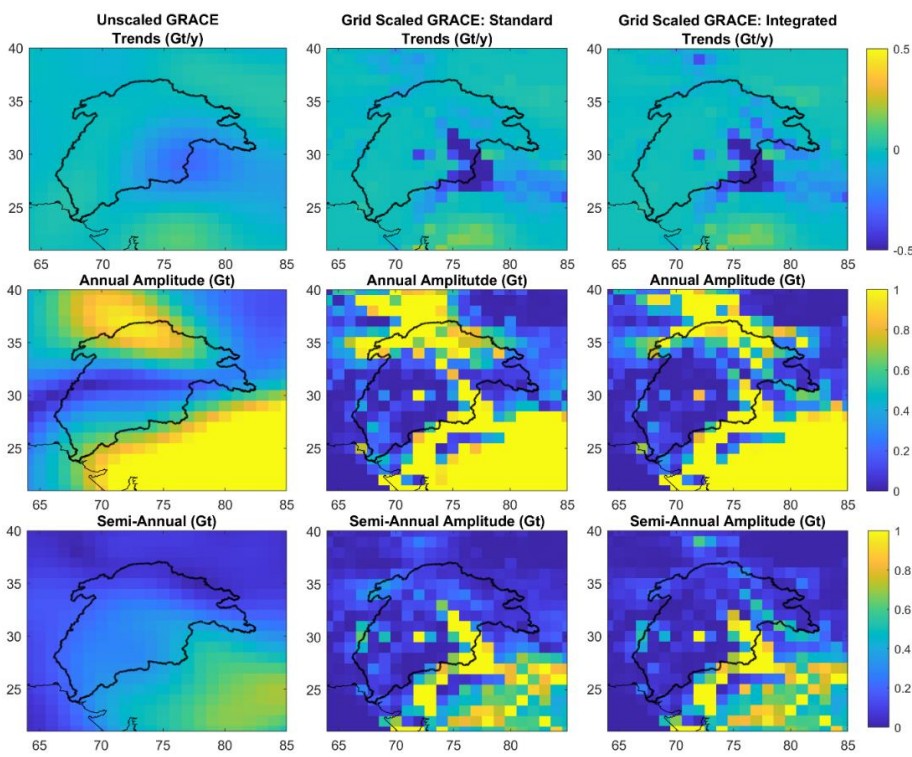

**Figure 13 Spatial distribution of trend, annual and semi-annual mass changes in unscaled GRACE (leftmost panel), grid scaled GRACE from standard version (middle panel), and grid scaled GRACE from Integrated version (rightmost panel) in Indus basin.**





Scaling based on the integrated model version leads to more negative trends and larger amplitudes than scaling based on the standard version, irrespective of the scaling scheme. Significant differences between scaling schemes occur in scaled semi-annual amplitudes. Grid scaling from the integrated version reduces the scaled semi-annual amplitude (compared to 23.8 Gt

from unscaled), while grid scaling from the standard version increases it. This is because pixels of significant semi-annual contribution (Fig. 7) lie in the non-glaciated region, where the scaling factors get smaller due to the addition of glacier mass changes (Fig. 12), leading to a reduction of semi-annual amplitude.

**Table 5 Parameters along with their formal uncertainties from time series fit to scaled GRACE SH estimates under different scaling**

**schemes**

| Scaling Schemes | Model Version | Trend (Gt/year) | Annual Amplitude (Gt) | Semi-Annual Amplitude (Gt) | RMSE (Gt) |
|---|---|---|---|---|---|
| Basin Scaling | Standard | $-8.7 \pm 0.6$ | $12.4 \pm 4$ | $28 \pm 4$ | 31.6 |
| | Integrated | $-9.2 \pm 0.7$ | $13.2 \pm 4$ | $30 \pm 4$ | 33.7 |
| Grid Scaling | Standard | $-11 \pm 0.6$ | $18.3 \pm 4$ | $25 \pm 4$ | 30.2 |
| | Integrated | $-12.1 \pm 0.6$ | $25 \pm 3$ | $21 \pm 3$ | 27.3 |
| Frequency Dependent Scaling | Standard | $-8.6 \pm 0.6$ | $16.8 \pm 3$ | $25.2 \pm 3$ | 27.7 |
| | Integrated | $-9.3 \pm 0.6$ | $17.2 \pm 3$ | $25.4 \pm 3$ | 27.7 |

A comparison of the grid-scaled GRACE SH time series to the MASCON time series (CSR-M) is shown in Fig. 14 to judge how realistic the values are. The $R^2$ values are similar to those obtained with unscaled GRACE SH, which provides confidence about the scaling factors obtained. We do not use this high value to judge the performance of scaling factors but only to

highlight their numerical authenticity. A similarly high $R^2$ for the de-trended time series shows that the correlation is not spurious.

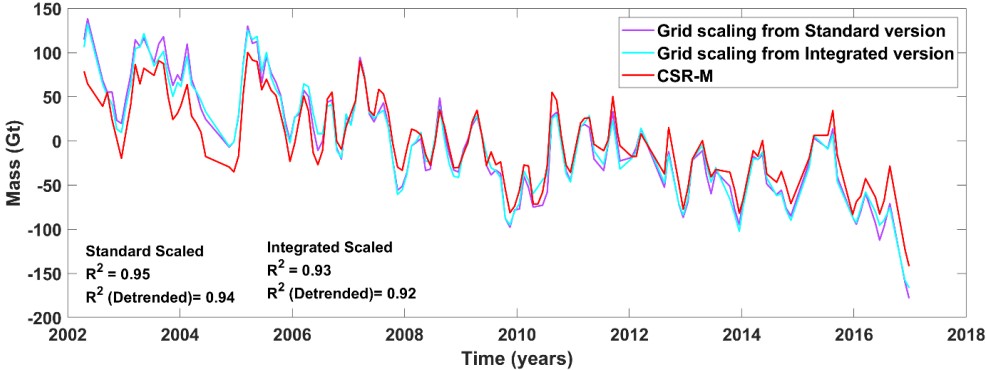

**Figure 14 Comparison of Grid scaled GRACE SH from Standard version and Integrated WGHM version with CSR-M.**

Similarly, the result of frequency-dependent scaling is compared with the MASCON time series in Fig. 15. The obtained $R^2$

values are higher than those of grid-scaling. This can be attributed primarily to the fact that the noise has not been scaled as in the grid scaling, making it more comparable to the MASCON time series. Even the real signals that may be present in the residual along with noise are not scaled. These real signals are mostly inter-annual, which are not much affected by filtering (sec. 3.2), and hence not scaling them leads to better correlation with the MASCON time series.





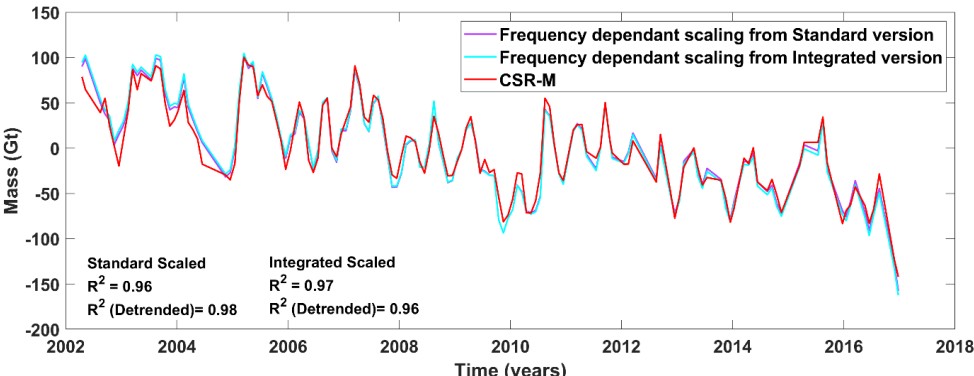

Figure 15 Comparison of Frequency-dependent scaled GRACE SH from Standard and Integrated WGHM versions with CSR-M.

### 3.5 Residual Leakage and Scaled Noise Estimates

Table 6 shows the initial leakage error and the noise level in unscaled GRACE estimates containing the random component of leakage. The initial leakage error determined from the standard version is less than that determined from the integrated version due to the smaller magnitude of TWS anomalies. The unscaled noise provides a baseline measure against scaled noise levels to evaluate the effect of scaling on random leakage components.

Table 6 Noise and Leakage Error in Unscaled GRACE estimates of Indus Basin

| Model Versions | Noise (Gt) | Initial Leakage Error (Gt) |
|---|---|---|
| Standard | 13.9 | 9.6 |
| Integrated | 13.9 | 11 |

Table 7 Scaled Noise and Residual Leakage in scaled GRACE estimates

| Scaling Scheme | Model Versions | Scaled Noise (Gt) | Residual Leakage Error (Gt) |
|---|---|---|---|
| Basin | Standard | 15.9 | 7.1 |
| | Integrated | 17 | 4.2 |
| Grid | Standard | 14.5 | 4.1 |
| | Integrated | 13.3 | 3.9 |
| Frequency-Dependent | Standard | 13.9 | 6.0 |
| | Integrated | 13.9 | 3.3 |

The scaled noise levels and the residual leakage errors from all three scaling schemes are shown in Table 7. These scaled noises implicitly contain the scaled random component of leakage, as explained in Sect. 2.3.3. Compared to unscaled GRACE



(Table 6), the basin and grid scaling causes the noise to be scaled along with the signal, whereas frequency-dependent scaling does not affect noise. Grid scaling leads to lower noise than basin scaling due to the suppression of random errors in most of the Indus basin regions where smaller scaling factors are present. Even smaller gridded scaling factors from the integrated version in those regions may explain the lower scaled noise. Basin scaling from the integrated version leads to higher noise than basin scaling from the standard version due to the higher magnitude of the scaling factor.

The residual leakage is lower than the initial leakage across all three schemes and model versions, indicating the effectiveness of the scaling factor approach. The residual leakage from the integrated version is lower than from the standard version across all schemes despite the larger initial leakage. Moreover, frequency-dependent scaling from the integrated version leads to the least residual leakage error. This again reinforces the inference that for basins like Indus, where mass changes of different frequencies are affected differently by filtering, frequency-dependent scaling performs best. However, for this scheme to perform best, adequate representation of the mass changes in the basin must be ensured by the model being used. Grid scaling from the integrated version seems to provide similar performance. This can be attributed to how different frequency mass changes are spatially localized in different basin regions, leading to similar behaviour (Fig. 7).

## 4 Summary and Conclusion

The study aimed to derive and evaluate scaling factors for the Indus River Basin from WGHM 2.2d using its Standard and Integrated versions, to account for the leakage effects in mass estimates derived from GRACE spherical harmonic solutions. Scaling factors were derived based on three schemes of different spatio-temporal characteristics: basin scaling factors that are spatially and temporally constant; gridded scaling factors that are spatially variable while temporally constant; and frequency-dependent scaling factors that are spatially constant and temporally variable. An evaluation was conducted in an inter-comparison framework that involved,

1. comparison of scaled time series with MASCON time series as a way to confirm the practicality of derived factors,
2. time-series analysis of scaled estimates to compare the effect of scaling on mass changes of different frequencies present in the time series and
3. comparison of noise and residual leakage in the scaled GRACE estimates.

The obtained results highlight the merits and demerits of each scaling scheme, which show that no scheme can be judged as the best for the region. The basin scaling scheme appears to be less sensitive to the model's mass distribution, which may justify the use of even worse models in their derivation. For the Indus basin, basin scaling factors seem to be driven by the filtering effect on the trend and may be used for applications dealing with long-term trends in the basin. However, a better signal-noise separation must be achieved to minimize the scaling of noise. Gridded scaling factors from the integrated version capture the spatial heterogeneity of water storage changes in the Indus basin. The fact that the addition of the glacier mass component modifies the scaling factors of the non-glaciated grid cells along with the glaciated grid cells is an interesting finding of the study. Since mass changes of different frequencies are localized in different regions of the Indus basin, the gridded scaling factors from both versions perform differently across different frequencies. The newly developed, frequency-dependent scale factors helped quantify the scaling effect on different temporal components of mass change. Frequency-dependent scaling shows that using a single basin scaling factor for basins like the Indus basin with mass changes occurring at different frequencies will lead to inappropriate scaling of one or more of these components. Moreover, frequency-dependent scaling keeps the noise levels unscaled, which can be extremely useful in applications requiring a high signal-to-noise ratio.



It is obvious to realize the possibility of a fourth scheme of gridded and frequency-dependent scaling that would provide three
scaling factors per grid cell. However, our initial experiments show that the time series decomposition at grid-scale leads to
vastly varying annual and semi-annual components and causes the scaling factors to be unrealistic. Hence, we leave that
development for future studies. The study is limited in providing an external validation with independent TWS estimates. The
usual approach is a comparison with TWS obtained from a water balance, but uncertainties in the derived TWS are much
larger than the changes due to scaling. Hence, we stick to an inter-comparison framework that can be realistically reproduced
and applied to different applications and studies that utilize GRACE data in the Indus basin and gain insights for choosing an
appropriate scaling scheme as per requirements. It may be stressed here that although the study has been done only for GRACE
data, it can naturally be extended to the ongoing GRACE-FO observations (with the availability of latest model outputs)
without any loss of generality.

**Code Availability**

The MATLAB scripts written to conduct this study can be obtained upon request to the corresponding author.

**Data Availability**

All GRACE Level 2 release 6 spherical harmonic datasets (NASA JPL, 2018; Dahle et al., 2018; UTCSR, 2018) used in this
study are available at http://grace.jpl.nasa.gov, supported by the NASA MEaSUREs Program. CSR release 6 Mascons were
downloaded from http://www2.csr.utexas.edu/grace (Save et al., 2016). The output of different WGHM model versions
(Cáceres et al., 2020) used in this study can be obtained upon request to the corresponding author. Maps of scaling factors
developed in this study can be obtained upon request to the corresponding author. Data required to reproduce all the time series
in the manuscript has been provided in the supplement. Sufficient information has been provided in the manuscript to reproduce
the results of this study using the datasets mentioned here.

**Supplement**

Data to reproduce all time series and periodogram plots in the manuscript has been provided in a separate supplementary file.
This can be used during the review process by the reviewers.

**Authors Contribution**

V.T. and A.G. designed the study, V.T. conducted the analysis and wrote the manuscript, M.H. supervised the GRACE data
processing and manuscript editing, R.R supervised the hydrological inferences and manuscript editing. All the authors provided
critical feedback, comments, and suggestions for the development of the manuscript.

**Competing Interests**

The authors declare that they have no known competing financial interests or personal relationships that could have appeared
to influence the work reported in this paper.




**Acknowledgment**

This study was conducted under the DAAD Kombinierte Studien- und Praxisaufenthalte für Ingenieure aus Entwicklungsländern (KOSPIE) Grant 2019 at Technische Universität Dresden, Institut für Planetare Geodäsie, Chair of Geodetic Earth System Research. We would like to thank Petra Döll and Denise Cáceres (Goethe University Frankfurt) for providing the WGHM v2.2d spatial grids of TWS anomalies and Benjamin D. Gutknecht (Technische Universität Dresden) for helping with their analysis.

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
