# Peer review of "Scaling methods of leakage correction in GRACE mass change estimates revisited for the complex hydro-climatic setting of the Indus basin"

_Hydrology and Earth System Sciences, 2022_

## Author Comment (AC1)

**Scaling methods of leakage correction in GRACE mass change estimates revisited for the complex hydro-climatic setting of the Indus basin**
V. Tripathi, A. Groh, M. Horwath, R. Ramsankaran

**Responses to reviewer 1**

**please see as well Decading Karakoram Anomaly**

**Reply:** We thank the reviewer for providing us an opportunity to highlight this interesting phenomenon. The 'Karakoram Anomaly' is termed as the stability or anomalous growth of glaciers in the central Karakoram, in contrast to the retreat of glaciers in other nearby mountainous ranges of Himalayas and other mountainous ranges of the world (Dimri, 2021). Various mass balances over this region have shown to be balanced or slightly positive (Farinotti et. al., 2020). There is, however, significant uncertainty over the reasons for this anomaly which seems to be caused by absence of reliable in-situ observations in this region.

However, GRACE observations of such small (nearly stable) trends will be contaminated with the nearby large negative trends due to leakage. This can be seen in figure 1, where the red bounding box demarcates the extent of Karakoram (Dimri, 2021). The scaled trends for these pixels lying within Indus basin (rightmost subfigure) are extremely small but negative (ranging from -0.01 to -0.1 Gt/y). The uncertainty associated with these values is however, nearly double. The two major limitations for the uncertainty of these values come from the inability of GRACE to resolve this extremely small signal from this phenomenon and the inability of underlying model to simulate this anomalous behavior at the current $1^0$ resolution, seen by large negative trends in the corresponding pixels (leftmost subfigure).

However, scaling does seem to make a distinction between pixels of Karakoram with less negative trends and the adjacent pixels of Ladakh region with more negative trends, which cannot be seen in unscaled GRACE fields (middle subfigure). We feel this to be a hopeful result indicating the presence of anomalous behavior in Karakoram, but further analysis of this phenomenon will be out of scope of this study.

[Figure]

Figure 1 Integrated WGHM, Unscaled GRACE and Scaled GRACE trend map in Gt/y at $1^0$ resolution. The red box indicates the extent of Karakoram where the anomaly is to be identified.

We will include this discussion in the revised version of the manuscript.

**References**

Dimri, A. P.: Decoding the Karakoram Anomaly, Science of The Total Environment, 788, 147864, https://doi.org/10.1016/j.scitotenv.2021.147864, 2021.

Farinotti, D., Immerzeel, W. W., de Kok, R. J., Quincey, D. J., and Dehecq, A.: Manifestations and mechanisms of the Karakoram glacier Anomaly, Nat. Geosci., 13, 8–16, https://doi.org/10.1038/s41561-019-0513-5, 2020.

---

## Author Comment (AC2)

**Scaling methods of leakage correction in GRACE mass change estimates revisited for the complex hydro-climatic setting of the Indus basin**

V. Tripathi, A. Groh, M. Horwath, R. Ramsankaran

**Responses to reviewer 2**

**Scaling methods of leakage correction in GRACE mass change estimates revisited for the complex hydro-climatic setting of the Indus basin written by Vasaw Tripathi and colleagues. The paper adresses the problem of spatial leakage in satellite gravimetry data which arises from the limited ability of sensor data from low-low satellite tracking to accurately resolve steep spatial gradients in surface mass anomalies at scales of a few hundred kilometers and smaller. The resulting systematic error (called spatial leakage) is often mitigated by means of scaling approaches, and the current article is following this avenue of research with a special emphasis on the Indus Basin. The paper is generally well written and complements the existing literature on applications of satellite gravimetry data in this catchment. The study moreover fits nicely into the scope of the journal so that acceptance might be recommended as soon as a number of concerns outlined below are properly addressed.**

**Reply:** We thank the reviewer for his positive and encouraging review, which we feel was instrumental in further improving the quality of our work. We address each concern individually as below,

**Major Comments**

**1. Rescaling (as performed in this study) is typically applied to allow for regional or even small-scale applications of the GRACE data. For the whole Indus catchment of about 1 million square kilometers, the effect of rescaling should be rather minor (which is also confirmed in the present work by the rather small changes in the time-series when moving from Figure 4 to Figures 14 and 15). It would be thus imperative to discuss in more detail the re-scaled GRACE-results presented in Figure 13, and also to compare them to the CSR mascons pixel by pixel. Note that I do not expect that rescaling will lead to a perfect match with the Mascons, nor do I suggest that the Mascons should be considered as the error-free truth in such an exercise. Instead, it would be important to carve out more explicitly any benefits of applying an elaborate rescaling scheme in lieu of simply downloading and applying the Mascons.**

**Reply:** We agree to the suggestion, following which we present figure 1 below, which includes the gridded trends, annual and semi-annual amplitudes from the CSR mascons (leftmost panel).

[Figure]

Figure 1 Spatial distributions of TWSA components from CSR Mascons, unscaled SH solutions, grid scaled SH solutions using standard and integrated WGHM models (left to right panels)

We can see that even though mascons restore some signal, they do it in such a way that signal is added mostly where signal already was in Spherical Harmonic (SH) solutions (notice the corresponding pixels in panel 1 and 2). Rescaling on the other hand, can be seen to offer much more spatial contrast, driven by spatial distribution of scaling factors. Therefore, if an ideal, perfect model were to be used then rescaling can allow downscaling of GRACE resolution along with leakage correction. For example, Mascons are unable to separate the glacier loss trends in the upper Indus pixels from GW depletion trends in pixels lying in southern part of Indus, while grid scaling from Integrated model does.

However, the tendency of gridded scaling factors to be driven by the dominant mass change component in the pixels (trends or seasonal), may lead to incorrect spatial patterns for one or more of these components. This will be addressed in detail in the reply to major comment 4.

The figure and discussion will be included in the revised version of the manuscript.

**2. A central result of this study are the scaling factors given in Figure 10. Table 4 gives a nice overview on the interpretation, which should be also reflected in the color scale selected for the figure. Please consider something like bright red or purple for all pixels with zero or even negative scaling, and neutral colors (green or white) for coefficients around one.**

**Reply:** We agree with the suggestion, following which we present figure 2 with an updated colour scale,

[Figure]

Figure 2 Scaling factors

**3. Data-driven methods to account for spatial leakage have been proposed in the past by some of the authors (e.g., 10.1002/2017WR021150), and such methods are now routinely applied to approximate spatial leakage in surface mass estimates obtained from GRACE spherical harmonics solutions as disseminated via gravis.gfz-potsdam.de. Since Gaussian filtering is readily available to all the authors of the present paper, it would be quite straightforward to additionally explore the usage of twice-filtered GRACE GSM fields for the assessment of spatial leakage in the Indus Catchment. I believe that such an additional experiment could nicely complement the existing material.**

**Reply:** We implemented the data driven correction (DDC) approach (Vishwakarma et.al., 2017) for the Indus basin and compared the results to our existing scaling approaches. The DDC method is a model independent leakage correction approach that uses twice filtered GRACE fields to determine and correct the leakage at basin scale. Therefore, we present the basin averaged time series from grid and frequency dependent scaling using integrated WGHM compared to the DDC time series in figure 3. Table 1 presents the corresponding time series components of leakage corrected basin averages from all three scaling schemes and the DDC method.

[Figure]

Figure 3 Basin averaged time series for Indus basin from DDC (green), Grid scaling (red) and frequency dependent scaling (blue) using integrated WGHM

Table 1 Parameters along with their formal uncertainties of the time series components of scaled GRACE SH estimates under different scaling schemes and the DDC method. The unscaled GRACE time series components are mention below for reference

| Scaling Schemes | Model Version | Trend (Gt/year) | Annual Amplitude (Gt) | Semi-Annual Amplitude (Gt) | RMSE (Gt) |
|---|---|---|---|---|---|
| Basin Scaling | Standard | -8.7 ± 0.6 | 12.4 ± 4 | 28 ± 4 | 31.6 |
| Basin Scaling | Integrated | -9.2 ± 0.7 | 13.2 ± 4 | 30 ± 4 | 33.7 |
| Grid Scaling | Standard | -11 ± 0.6 | 18.3 ± 4 | 25 ± 4 | 30.2 |
| Grid Scaling | Integrated | -12.1 ± 0.6 | 25 ± 3 | 21 ± 3 | 27.3 |
| Frequency Dependent Scaling | Standard | -8.6 ± 0.6 | 16.8 ± 3 | 25.2 ± 3 | 27.7 |
| Frequency Dependent Scaling | Integrated | -9.3 ± 0.6 | 17.2 ± 3 | 25.4 ± 3 | 27.7 |
| DDC | - | -8±0.6 | 16.3±3.6 | 26.4±3.6 | 30.3 |
| Unscaled GRACE | - | -7.6±0.5 | 11.8±3.3 | 24.8±3.3 | 27.7 |

*\* The annual and semi-annual amplitudes from unscaled GRACE in table 2 in the preprint were changed slightly due to a reporting error. This change does not affect the conclusions from these results.*

It can be seen from figure 3 that overall, frequency dependent scaling using WGHM agrees extremely well with the independent DDC method for the Indus basin. The agreement is also depicted in the time series components in table 1. The small differences are well within the uncertainty limits and can be attributed to the methods being entirely different. Larger differences are however, seen between grid scaling and DDC, which highlight the limitation of grid scaling to over or under scale certain pixels. Grid scaling can be seen to overestimate the trends compared to DDC, since the trend contributing pixels get scaled with larger scaling factors. Thus, this effect is seen to be more pronounced in grid scaling from integrated version than from standard version. The seasonal amplitudes have large differences but their nature cannot be generalized between annual and semi-annual frequencies. This highlights the differences in physical processes governing the large scale and fine scale behaviour of these components. Therefore, the results provide confidence in the ability of the new frequency dependent scaling using WGHM to correctly restore the damaged signal at catchment scale.

We however, refrain from using this inter-comparison to establish a 'best' scaling method, since the DDC method is reported to have limitations in heterogeneous basin like Indus, arising from the approximations inherent to the method, that are more applicable to catchments surrounded by catchments with similar hydrological activities (Vishwakarma et. al., 2017).

The figures and the discussion will be included in the revised version of the manuscript.

**4. I understand that spatially distributed in situ data is not readily available in a transboundary basin like the Indus catchment. In such a situation, a common approach to demonstrate the applicability of a new method developments would be the usage of simulated sensor data and satellite products, where the true mass variability entering the simulations is known (see, e.g., 10.1007/s10712-015-9338-y). In such a simulation environment, it could be demonstrated to what extent rescaling as proposed here mitigates the adverse effects of spatial leakage.**

**Reply:** We are very thankful to the reviewer for suggesting such a simulation experiment, which offered us newer insights into our work and the scaling method in general.

We set up a simulation environment, using the WGHM fields as a proxy for true TWS anomalies. We corrupted these fields with GRACE like noise. For this, we first derived the error covariance matrices from the normal equations provided by TU Graz for the ITSG solutions (Kvas et. al., 2019). Using Cholesky decomposition and normally distributed random numbers, we generated random realizations of GRACE errors in the SH domain. We then added these errors to the WGHM fields in SH domain and filtered using the same filter as used in the study for GRACE (Swenson destriping + 300 km Gaussian). These filtered and corrupted WGHM fields now represent GRACE like observations. Then using the scaling factors derived in the study, we rescaled these filtered and corrupted WGHM fields to recover the lost signals.

The results of the simulation are shown in form of time series components of the rescaled WGHM fields compared to the original true values in table 2 for standard version and table 3 for the integrated version. The rescaled spatial fields from grid scaling compared to true fields are shown in figure 4 for the standard version and figure 5 for the integrated version.

Table 2 Time series components of basin averages from the simulation using standard WGHM

| Simulation results with Standard WGHM | | | | |
|---|---|---|---|---|
| | Trend | Annual | Semi-Annual | RMS |
| True Standard | -10.2±0.4 | 17.4±2.3 | 18±2.3 | 19.2 |
| Filtered Corrupted Standard | -8.6±0.4 | 10.7±2.1 | 17.6±2.1 | 17.4 |
| Basin Scaled | -9.8±0.4 | 12.2±2.4 | 20±2.4 | 19.9 |
| Grid Scaled | -11.4±0.4 | 12.2±2.4 | 19.9±2.4 | 19.9 |
| Frequency Scaled | -9.8±0.4 | 17.6±2.1 | 18.1±2.1 | 17.4 |

Table 3 Time series components of basin averages from the simulation using integrated WGHM

| Simulation results with Integrated WGHM | | | | |
|---|---|---|---|---|
| | Trend | Annual | Semi-Annual | RMS |
| True Integrated | -20.4±0.4 | 19.1±2.2 | 18±2.2 | 18.3 |
| Filtered Corrupted Integrated | -16.3±0.4 | 12.3±2 | 17.5±2 | 17 |
| Basin Scaled | -19.8±0.4 | 15±2.5 | 21.3±2.5 | 20.7 |
| Grid Scaled | -21.2±0.4 | 17.4±2.2 | 17.1±2.2 | 18.4 |
| Frequency Scaled | -20±0.4 | 19.8±2 | 18.2±2 | 17 |

[Figure]

Figure 4 True (leftmost panel) and recovered (rightmost panel) standard WGHM fields from grid scaling simulation

[Figure]

Figure 5 True (leftmost panel) and recovered (rightmost panel) integrated WGHM fields from grid scaling simulation

The simulation experiments establish the frequency dependent scaling as the best performing scheme in terms of recovering the true basin averaged signal for both the model versions. The similarity of recovered trends from basin and frequency dependent scaling, support the inference that the basin scaling factors are driven by the dominant mass change component, which is the trend in case of Indus basin. Recovered trends from grid scaling are however overestimated compared to the true trends, which support the inference made for GRACE observations in the current manuscript (lines 455-457). The recovered semi-annual amplitude from grid scaling using integrated version is reduced while using the standard version, is increased (compared to the filtered corrupted semi-annual component). Most of the semi-annual signal contribution comes from non-glaciated pixels in the trunk Indus (fig. 4 and 5, rightmost panel). Upon addition of glacier component in the standard model, the scaling factors for these pixels get smaller and hence the semi-annual amplitude is decreased. The greater number of such pixels outweighs the increase in semi-annual signal of few pixels in the upper Indus (fig. 4, leftmost panel). This again supports the inference made for GRACE estimates in the current manuscript (lines 468-472).

It can be seen that grid scaling is unable to recover the true annual signal in both model versions, falling short by 29% in case of standard and 8% in case of integrated versions. In the current manuscript, however, we made an inference that grid scaling overestimates the annual amplitude compared to frequency dependent scaling (lines 458-461), which is contradicted by the simulation results. In order to explain this contradiction, we break it down to two separate questions. Why is the recovered annual amplitude less than the true annual amplitude (or equivalently, to annual amplitude recovered by frequency dependent scaling) in case of simulations? Why is then the rescaled GRACE annual amplitude from grid scaling more than from frequency dependent scaling (table 1)?

Firstly, in figure 4 and 5, corresponding to the fields of annual amplitude, it can be seen that the scaling is unable to recover the true spatial pattern. Most of annual amplitude coming from pixels along the Indus River channel (in Trunk Indus) is lost due to small scaling factors. Although, few pixels in the upper Indus basin contributing to annual amplitude (fig. 4 and 5, leftmost panel), get over scaled due to larger scaling factors,

the net compensating effect is loss of amplitude due to smaller number of such pixels. This explains why the recovered amplitude falls short of the true value.

The second question can be explained by comparing the field of unscaled GRACE annual amplitude in figure 1 with the field of annual amplitude in filtered corrupted standard model in figure 4. The pixels from GRACE in the upper Indus basin already hold much stronger annual signal compared to the corresponding pixels from the filtered corrupted model. Therefore, the effect of larger scaling factors for such pixels, make the overall rescaled annual amplitude to be larger than rescaled annual amplitude from frequency dependent scaling. For the simulation, this effect of larger scaling factors is compensated by the already weaker annual amplitude to be scaled, leading to lower contribution to the overall annual amplitude which falls short of the true value.

Therefore, we think that we were too superficial in making the above inference in the current manuscript (lines 458-463) and will replace it with the discussion above in the revised version.

Following the simulations, we re-calculated the initial and residual leakages (as in table 6 and 7 of current manuscript) using the more realistic filtered and corrupted model in equations 11 and 12, instead of just filtered model. The new estimates are shown in table 4 and 5 below. The residual leakages are least in frequency dependent scaling, indicating its better performance. Residual leakages from grid scaling are indicative of their inability to reproduce the spatial pattern of seasonal signals in the basin. We must caution here, that although the residual leakage from grid scaling using integrated version is lower, it is only because of small magnitude of scaling factors. The underlying spatial patterns are no better recovered compared to grid scaling from the standard model.

Table 4 Initial leakage error

| Model Version | Initial Leakage Error (Gt) |
|---|---|
| Standard | 10.7 |
| Integrated | 12 |

Table 5 Residual Leakage Error

| Scaling Scheme | Model Version | Residual Leakage (Gt) |
|---|---|---|
| Basin | Standard | 7.9 |
| | Integrated | 5 |
| Grid | Standard | 11 |
| | Integrated | 5.6 |
| Frequency-Dependent | Standard | 6.7 |
| | Integrated | 4 |

**5. I am not quite convinced that frequency-dependent scaling (as proposed here) will be "useful for applications requiring a high signal-to-noise ratio". After all, it is any deviation from the seasonal cycle related to either interannual climate variations or hydrometeorological extremes that is of particular interest in many applications of remote sensing data. I do not find arguments in the paper that would help restoring leakage for such signals, so please elaborate your claim a little further.**

**Reply:** The applications that we refer here are those that require accurate knowledge of seasonal cycle. Such applications include water availability studies by decision making bodies to ensure safe supply of water to an area every year. In such applications, frequency dependent scaling will be a robust scaling method, that could

provide correct seasonal signals without scaling the underlying noise itself, thus giving more confidence in the practical usage of such estimates.

However, we agree that other than seasonal cycle, studies of interannual variations and occurrence of hydrological extremes are of immense interest to the community, both scientific and decision making. The proposed frequency dependent scaling will not be useful for applications that require accurate estimates of such signals. For such signals, specific scaling schemes would be required. We agree to keep the possibility of such analysis as a future extension of existing scaling methods, but currently out of scope of this study.

An interesting observation can however be made with regards to the characteristics of such signals. Such signals which can be defined as deviation from the seasonal cycle, will definitely require a-priori accurate seasonal signal with low noise, which can be provided by frequency dependent scaling. The second step would then proceed to scale the deviations appropriately. Hence frequency dependent scaling may be indirectly useful, even in such cases.

We will include update and include this discussion in the revised manuscript.

**Minor Comments**

**line 29: It is surprising to find a PhD thesis cited for such a rather general statement. In case you would like to give credit to the work of this author specifically, please consider citing any of her research papers.**

**Reply:** Agreed and will be changed in revision

**Line 44: Authors should understand that even perfect dealiasing models will not remove all spatially correlated errors. On the contrary, simulation studies (10.1007/s10712-015-9338-y) demonstrate that background model errors in tides and sensor noise of the GRACE accelerometers have an almost similarly large effect on accuracy and spatial resolution of the GRACE monthly solutions available today. This also applies to GRACE-FO, where one of the accelerometer instruments performs worse than expected.**

**Reply:** We mentioned a few lines earlier (lines 37-39) that the "limited spatial resolution is related to errors arising from the measurement process of GRACE and the modelling deficiencies in the estimation of gravity field parameters". We agree and the description in the following lines distract from the measurement errors as a major source of gravity field error. We will revise this passage, possibly making it more concise (as this manuscript is not about explaining error in GRACE solutions) and providing more references.

**Line 49: Spatial leakage errors do not occur from the truncation of the spherical harmonics expansion but from the limited resolution of the along-track sensor data and the upward continuation of the gravity field from the surface to the orbital height of the satellites. Further, there is some inherent smoothing in the sensor data pre-processing that also reduces spatial gradients in the resulting gravity field estimates. Expanding the cutoff degree will certainly not solve the leakage issues, but render the inversion problem ill-posed.**

**Reply:** By truncation we rather meant the inevitable limitation of the spherical harmonic degree to which the GRACE solutions can be reasonably provided. We will change the misleading wording in the revised manuscript.

**Line 97: dependent**

**Reply:** Changed

**Line 104: Incomplete sentence. I assume that those glaciers are additionally feeding the Indus via tributary rivers?**

**Reply:** Yes, the sentence was incomplete. It will be updated in revised manuscript.

**Table 1: I am not convinced that "Integrated WGHM" is a very intuitive name for this model experiment. What about WGHM+GGM?**

**Reply:** The names of the model version are the same as given by the modelers (Cáceres et al., 2020). We adopt this so that readers who are already aware of the Integrated WGHM, can directly recall what it is and readers who are not aware of it, can refer to the model literature with the same parent name to familiarize themselves.

Although we explain it in the text (lines 86-87), we do note the confusion that can arise among the readers who just reference the table 1. Hence, we will add a short description in the title for the table indicating Integrated version is in fact WGHM integrated with GGM.

**Line 249: In Table 1, eight different model experiments are listed: Please explain more explicitly why just two of them are applied in the following.**

**Reply:** In lines 150-157, we have explained in detail what the eight model versions are and why two (average of four variants under each version; standard and integrated) were used.

However, we understand that just looking at the table, such a doubt would arise. Therefore, we will modify the title of the table to explicitly indicate that the two versions used are the average of four variants under each model version and it is done since no single variant can adequately represent the heterogenous hydroclimatic conditions of Indus basin.

**References**

Vishwakarma, B. D., Horwath, M., Devaraju, B., Groh, A., and Sneeuw, N.: A Data-Driven Approach for Repairing the Hydrological Catchment Signal Damage Due to Filtering of GRACE Products: REPAIRING SIGNAL DAMAGE DUE TO FILTERING, Water Resour. Res., 53, 9824–9844, https://doi.org/10.1002/2017WR021150, 2017.

Kvas, A., Behzadpour, S., Ellmer, M., Klinger, B., Strasser, S., Zehentner, N., and Mayer-Gürr, T.: ITSG-Grace2018: Overview and Evaluation of a New GRACE-Only Gravity Field Time Series, J. Geophys. Res. Solid Earth, 124, 9332–9344, https://doi.org/10.1029/2019JB017415, 2019.

Cáceres, D., Marzeion, B., Malles, J. H., Gutknecht, B. D., Müller Schmied, H., and Döll, P.: Assessing global water mass transfers from continents to oceans over the period 1948–2016, Hydrol. Earth Syst. Sci., 24, 4831–4851, https://doi.org/10.5194/hess-24-4831-2020, 2020.